# High salt exacerbates acute kidney injury by disturbing the activation of CD5L/apoptosis inhibitor of macrophage (AIM) protein

Ching-Ting Wang[1], Tetsushi Tezuka[1], Naoki Takeda[2], Kimi Araki[2], Satoko Arai[1,3]*, Toru Miyazaki[1,3,4,5]*

1 Laboratory of Molecular Biomedicine for Pathogenesis, Center for Disease Biology and Integrative Medicine, Faculty of Medicine, The University of Tokyo, Tokyo, Japan, 2 Institute of Resource Development and Analysis, Division of Developmental Genetics, Kumamoto University, Kumamoto, Japan, 3 The Institute for AIM Medicine, Tokyo, Japan, 4 LEAP, Japan Agency for Medical Research and Development, Tokyo, Japan, 5 Laboratoire d'ImmunoRhumatologie Moléculaire, Plateforme GENOMAX, Institut National de la Santé et de la Recherche Médicale UMR_S 1109, Faculté de Médecine, Fédération Hospitalo-Universitaire OMICARE, Fédération de Médecine Translationnelle de Strasbourg, Laboratory of Excellence TRANSPLANTEX, Université de Strasbourg, Strasbourg, France

* tm@m.u-tokyo.ac.jp (TM); sarai@m.u-tokyo.ac.jp (SA)

**Data Availability Statement:** All relevant data are within the manuscript and its Supporting Information files.

## Abstract

The influence of excess salt intake on acute kidney injury (AKI) has not been examined precisely except for some clinical data, unlike in chronic kidney disease. Here, we addressed the influence of high salt (HS) on AKI and its underlying mechanisms in terms of the activity of circulating apoptosis inhibitor of macrophage (AIM, also called CD5L) protein, a facilitator of AKI repair. HS loading in mice subjected to ischemia/reperfusion (IR) resulted in high mortality with advanced renal tubular obstruction and marked exacerbation in biomarkers of proximal renal tubular damage. This AKI exacerbation appeared to be caused mainly by the reduced AIM dissociation from IgM pentamer in serum, as IgM-free AIM is indispensable for the removal of intratubular debris to facilitate AKI repair. Injection of recombinant AIM (rAIM) ameliorated the AKI induced by IR/HS, dramatically improving the tubular damage and mouse survival. The repair of lethal AKI by AIM was dependent on AIM/ kidney injury molecule-1 (KIM-1) axis, as rAIM injection was not effective in KIM-1 deficient mice. Our results demonstrate that the inhibition of AIM dissociation from IgM is an important reason for the exacerbation of AKI by HS, that AIM is a strong therapeutic tool for severe AKI.

## Introduction

There is a well-accepted consensus that excess salt (i.e., NaCl) intake has detrimental physiological effects, resulting in different outcomes such as hypertension, kidney disease, and cardiovascular disease [1]. Numerous clinical studies have demonstrated that a reduction of dietary salt intake has clear kidney protective effects, including a decrease in systolic blood pressure and proteinuria [2–8]. Based on the evidence, the international guidelines for the treatment of chronic kidney disease (CKD) recommend a daily intake of NaCl <5 g, which is

**Funding:** This work was supported by: - CREST (grant number JP18gm0610009) funded by Japan Agency for Medical Research Development to Toru Miyazaki; - LEAP (grant number JP21gm0010006), funded by Japan Agency for Medical Research Development to Toru Miyazaki; - Grant-in-Aid for Scientific Research (B) grant number 16H05313, funded by Ministry of Education, Culture, Sports, Science and Technology to Satoko Arai. The funders had no role in study design, data collection and analysis, decision to publish, or preparation of the manuscript.

**Competing interests:** The authors have declared that no competing interests exist.

equivalent to 2 g sodium. Although the precise mechanisms underlying the pathogenic effects of high salt loading on kidney tissue are unclear, recent studies have shown the involvement of multiple microenvironmental changes, including chronic inflammation and fibrosis as well as tissue remodeling [9–13], in addition to the change in body fluid volume, which was originally suggested as the biggest impact of salt [14, 15]. However, high salt loading is considered to have a chronic effect, and thus its deleterious effects have mostly been studied on CKD and not on acute kidney injury (AKI). Few studies, particularly in terms of mechanisms using animal models, have been performed to address the effects of high salt loading on AKI.

AKI is characterized by impaired kidney filtration function, caused by different insults such as kidney ischemia/reperfusion (IR) injury, sepsis, or nephrotoxins. Even AKI is often reversible, it has been associated with high morbidity and mortality. The incidence of AKI has been rising, but so far, no effective clinical therapies have been developed [16–18]. One of the most important pathological characteristics broadly observed in AKI is massive cell death in renal tubular epithelial cells, which leads to luminal obstruction by dead cell debris, particularly in the proximal tubules at the corticomedullary junction [19–21]. Sustained tubular obstruction results in a marked reduction of glomerular filtration and the continuous production of inflammatory mediators by intraluminal dead cells. These events further exacerbate tubular injury, often leading to the death of patients due to renal failure [19]. Thus, in AKI, it is apparent that the rapid and efficient removal of luminal dead cell debris is required to restore tubular structure and renal function as well as to improve the survival of patients [22–26]. Previously, we demonstrated that intravenous injection of mice with apoptosis inhibitor of macrophage (AIM; also called CD5 antigen-like [CD5L]) protein successfully repaired AKI by promoting the removal of intraluminal dead cell debris [27].

AIM is a circulating protein that was initially identified as a supporter of macrophage survival [28] and is now recognized as a facilitator of repair in many diseases including AKI [27], post-dialysis chronic peritonitis [29], multiple sclerosis [30], liver cancer [31], fatty liver, and obesity [31, 32]. Of the various mechanisms exerted by AIM, enhancement of the phagocytic removal of dead cells was particularly highlighted during the repair of AKI [27]. AIM consists of three cysteine-rich domains (termed SRCR domains), and a unique positively charged amino acid cluster is present at the carboxyl terminal within the third SRCR domain [28]. This cluster forms a charge-based interaction with dead cells, whose surface is strongly negatively charged due to the high exposure of phosphatidylserine [27]. This association markedly enhances the engulfment of dead cells by phagocytes based on the nature of AIM; that is, AIM is highly internalized by phagocytes via multiple scavenger receptors [33]. At the lumen of proximal tubules under AKI, the phagocytes are injured, but surviving tubular epithelial cells highly express kidney injury molecule-1 (KIM-1), a scavenger receptor that functions as a ligand of AIM [27]. Interestingly, although AIM associates with the large IgM pentamer in serum, which protects AIM from renal excretion, AIM dissociates from IgM during AKI and reaches the intraluminal debris to facilitate AKI repair [27, 34, 35].

We addressed two major specific questions in this study. First, we assessed the influence of high salt on AKI and its underlying mechanisms. Second, we tested whether AIM was therapeutically effective for AKI under high salt loading. To answer these questions, we performed various experiments using animal models of AKI.

## Materials and methods

### Mice

CD1 (ICR) mice were purchased from Charles River Laboratories Japan (Kanagawa, Japan). C57BL/6 (B6) mice were purchased from CLEA Japan (Tokyo, Japan). All mice were

maintained under specific-pathogen-free conditions and all animal experiments were conducted in strict accordance with the recommendations in the Guide for the Care and Use of Laboratory Animals of the National Institutes of Health. The protocol was approved by the Committee on the Ethics of Animal Experiments of the University of Tokyo (permission number: P15-126). All surgeries were performed under inhalational isoflurane anesthesia and analgia, and all efforts were made to limit discomfort for mice such as warmth, soft bedding and assisted feeding. Our manuscript confirming the study is reported in accordance with ARRIVE guidelines.

## Gene knockout by CRISPR/Cas9

To generate $AIM^{-/-}$ mice on a CD1 background, two types of pX335 vectors (Addgene, Watertown, MA) carrying 5′-caccgaacaatggagccatggcc-3′ or 5′-caccggtgagtgtccctgcttctg-3′ were microinjected into the prenuclei of fertilized eggs of CD1 mice. Thereafter, the two-cell embryos were transferred into the uterus of pseudo-pregnant female mice. Genomic DNA isolated from the tail of progenies was tested for any deletion within the *cd5l* (*AIM*) gene locus by PCR and sequencing. An $AIM^{-/-}$ mouse line carrying an appropriate deletion at the first ATG region was expanded and used for experiments. $KIM\text{-}1^{-/-}$ mice were created along a similar strategy by injecting pX335 vectors carrying 5′-caccgaagacttgaatctgattca-3′ or 5′-caccggcctca-tactgcttctccc-3′ into the fertilized eggs of B6 mice. The validation of the newly generated $AIM^{-/-}$ CD1 mice and the $KIM\text{-}1^{-/-}$ mice is shown in S1 Fig.

## IR and high salt loading

IR was conducted as described previously [36]. Briefly, 7–9-week-old mice underwent unilateral right nephrectomy 1 week before IR. For IR, the mice were anesthetized by isoflurane to achieve analgesia, and the left kidney was exposed and the renal artery and veins were occluded with clamps for 15 min for B6-strain mice and 18.5 min for ICR-strain mice. The mice were maintained at 37°C on a warming plate during the entire ischemic period. We frequently observed the skin warmness and the mouse status to keep post-IR mice warm. After the ischemic process, the clamps were removed to provoke blood reperfusion and the mice were fed regular chow with 1.0% NaCl-containing water or normal water. For intraperitoneal high salt loading, 5 mL of 5% dextrose in water or 0.9% saline was injected intraperitoneally into the mice every day until the end point. Furthermore, we strictly complied with the requirement of a humane endpoint. We carefully observed the post-IR mice and euthanized the individual mouse and used for analysis, if either significant decrease in renal function (serum creatinine level 3.0 or higher) due to progression of renal damage, difficulty in feeding/water intake, symptoms of agony, long-term appearance abnormalities with no signs of recovery, rapid weight loss, was shown.

## Antibodies and reagents

Antibodies and reagents used for histological and biochemical experiments are as follows. Primary antibodies specific for: AIM (Rab1 rabbit polyclonal for mice western blot; #35 and #36 (for mouse urinary AIM) were established in our laboratory, Kim-1 (rat polyclonal for mice IHC, R&D Systems). Secondary antibodies and related reagents are: goat anti-rabbit IgG HRP (Invitrogen), streptavidin HRP (BD), G-block (Genostaff) and Simple Stain Rat MAX PO (NICHIREI BIOSCIENCES). Antibodies for flowcytometry are: Brilliant Violet 650 rat anti-mouse CD45 Antibody (30-F11, BioLegend), BV605 Rat anti-mouse Ly-6G (RB6-8C5, BD), BV510 rat anti-CD11b (M1/70, BD), BV421 rat anti-mouse F4/80 (T45-2342, BD), APC rat anti-mouse TNFα (MP6-XT22, Thermo Fisher Scientific), PE rat anti-mouse IL-6 (MP5-20F3,

BD), APC-eFluor 780 rat anti-mouse IL-1β (Pro-form) (NJTEN3, Thermo Fisher Scientific), APC-eFluor 780, Thermo Fisher Scientific), FITC rat anti-mouse CCL2 (MCP-1) (2H5, Thermo Fisher Scientific), Purified rat anti-mouse CD16/CD32 (2.4G2, Mouse BD Fc Block).

## Serum biomarkers

Serum Cre concentrations were measured using a Lab-Assay Creatinine Kit (Wako Pure Chemical Co., Ltd., Osaka, Japan). Serum BUN levels were determined using the FUJI DRI-CHEM 4000 V analyzer system (FUJIFILM Co., Ltd., Tokyo, Japan). Serum and urine sodium/chloride levels were analyzed by Oriental Yeast Co., Ltd. (Tokyo, Japan). Serum IS levels were measured by FUSHIMI Pharmaceutical Co., Ltd. (Kagawa, Japan).

## Histology and ATN score

The kidneys were fixed in 4% paraformaldehyde in phosphate-buffered saline (PBS) for 24 h and embedded in paraffin. PAS staining was performed on 4-μm sections of paraffin-embedded kidney blocks to evaluate renal damage. PAS-stained kidney specimens were used to evaluate renal destruction. More than 8 sections of the kidney from at least 4 different mice of each group were examined. The ATN score was graded according to the percentage of abnormal proximal tubules (proximal tubule dilation, brush border damage, proteinaceous casts, interstitial widening, and necrosis) in one selected region of the kidney (0, none; 1, <10%; 2, 10%–25%; 3, 26%–45%; 4, 46%–75%; 5, >75%) as described previously [27]. Four regions in one kidney were investigated in each mouse.

## Hemodynamics

Total renal blood flow was measured by a laser blood flowmeter (OMEGAFLO-Lab; OMEGA-WAVE, Inc., Tokyo, Japan). The mice were anesthetized with inhalational isoflurane and the probe was attached lightly to the surface of the exposed kidney. Blood flow values were calculated as the average of continuous dynamic measurement for 5 s. Data are presented as the percentage decrease of blood flow compared with the pre-IR condition. For blood pressure investigation, the mice were fixed in an animal holder and the systolic blood pressure of tail arteries was measured by a non-invasive tail-cuff system (Muromachi Kikai Co., Ltd., Tokyo, Japan). Three measurements were conducted for each mouse on days 0 to 3.

## Quantitative PCR

The quantitative evaluation of mRNA levels was performed by the SYBR Green-based ΔΔCT method using a QuantStudio 3 Real-Time PCR system (Invitrogen, CA). Sequences of the oligonucleotides used are presented in S1 Table.

## Dissociation of kidney cells and flowcytometric analysis

Dissociation of kidney cells was performed as previously described [27]. The kidney was rinsed with Hank's Balanced Salt Solution (HBSS; ThermoFisher Scientific) and the samples were roughly minced with scissors and kept in HBSS on ice. For digestion, the gentleMACS dissociator (Miltenyi Biotec) was used with the Mouse Tumor Dissociation Kit (Miltenyi Biotec), using HBSS containing 1% BSA. The tissue was incubated in the enzyme mixture for 30 min at 37˚C after homogenizing with gentleMACS program 'm_brain_01'. The gentleMACS program 'm_intestine_01' was applied for the final homogenization. After inactivating the enzymes with autoMACS running buffer (Miltenyi Biotec), the cell suspension was passed through 70-μm EASYstrainer (Greiner Bio-One).

Before staining the surface molecules of dissociated kidney cells, Fc receptors were blocked with anti-mouse CD16/32. Then cells were stained for CD45, CD11b, Ly-6G and F4/80 for the surface expression. Thereafter, TNFα, IL-6, IL-1β and MCP-1 were stained intracellularly using Cytofix/Cytoperm Fixation/Permeablization Kit (BD). F4/80-high or low cells in $CD45^+Ly$-$6G^-CD11b^+$ cells were analyzed for the intracellular expression of pro-inflammatory factors by flowcytometory (FACSCelesta, BD).

### ELISA for urinary AIM

All ELISA assays were performed in duplicate. Mouse urinary AIM was measured by ELISA using two different rat anti–mouse AIM monoclonal antibodies (rat IgG, clones #36 and #35; gener¬ated in our laboratory). The lower limit of quantification, as assessed by using recom¬binant AIM protein as a standard, was 0.0625 ng/ml for mouse AIM. To analyze mouse AIM, urine samples were diluted at 1:10 to 1:50.

### Food and water intake

On day 1 after I/R surgery, all the mice were placed in the metabolic cages and given a pre-weighed amount of food and water, and the intake was recorded every 24 hours.

### Generation and purification of rAIM protein

Chinese hamster ovary cells were transfected with pcDNA3.1-rAIM plasmid and cultured in CD FortiCHO medium (Invitrogen) for 3 days. rAIM was purified from the culture superna-tant using an in-house-made rat anti-mouse AIM monoclonal antibody (clone 36) conjugated to Protein G–Sepharose (GE Healthcare Life Sciences, PA). Bound protein was eluted with 0.1 M glycine-HCl (pH 3.0) and neutralized with 1 M Tris-HCl (pH 8.5). Protein was concen-trated as necessary using Amicon Ultra filter concentrators (Millipore, MA) and stored at −80˚C in PBS. Endotoxin levels were measured by the chromogenic Limulus amebocyte lysate endotoxin detection system (GenScript, NJ) following the manufacturer's protocols. Protein concentration was determined by the bicinchoninic acid assay according to the manufacturer's protocol (Pierce, IL).

### Statistical analysis

Data were analyzed using EZR software or BellCurve for Excel (Social Survey Research Infor-mation Co., Ltd.) and are presented as mean values ± s. d. unless otherwise specified. Paired results were assessed using parametric tests such as Welch's $t$-test. Comparisons among multi-ple groups were analyzed using one-way ANOVA followed by the Holm's *post hoc* test. Com-parisons between two groups at different time points were analyzed using Welch's $t$-test at each time point followed by Holm's adjustment of $P$ values. For two-factor comparison, two-way ANOVA was performed followed by multiple comparison test using Tukey's *post hoc* test. For Kaplan-Meier curves, $P$ values were determined with the log-rank test. Error bar: mean ± s. d. * $P < 0.05$, ** $P < 0.01$, and *** $P < 0.001$, unless specified.

## Results

### High salt exacerbates AKI in mice and increases their mortality

To assess the influence of high salt loading on AKI progression in an animal model, CD1 mice that had undergone kidney IR following single nephrectomy were fed high salt (HS; 1.0% NaCl) in drinking water immediately after IR surgery. We used CD1 mice because this strain is sensitive to IR [37]. Remarkably, nearly 80% of HS-fed mice died after day 4 post-IR,

whereas the mortality of IR mice without the HS was only 25% by day 7 (Fig 1A). Histologic analysis of HS-fed mice at day 3 post-IR revealed the massive obstruction of renal tubules by Periodic acid–Schiff (PAS)-stained dead cell debris, mainly at the medulla-cortex border, but also across a broad area of the outer cortex (Fig 1B). The acute tubular necrosis (ATN) score was worse in HS-fed IR mice than in IR mice without HS (Fig 1C). Quantitative PCR analysis detected significantly higher mRNA levels of the representative tubular injury markers *Havcr1* (*KIM-1*) and *Ngal* in IR HS-fed mice (Fig 1D). Intraperitoneal injection of 0.9% NaCl solution instead of the HS diet accelerated the AKI symptoms similarly in terms of survival and histologic outlook (Fig 1E). Thus, high salt loading, which is known to accelerate CKD, is also detrimental to AKI pathology and has immediate effects.

## High salt does not accelerate acute renal failure but worsens the uremic state

Intriguingly, the increase in serum creatinine (Cre) and blood urea nitrogen (BUN) levels in HS-fed IR mice did not correlate with the severity of kidney damage judged by histology and mortality until day 5: their serum levels were almost comparable with those in IR mice without HS (Fig 2A). This result suggests that HS exacerbated tubular damage rather than glomerular injury, and thus did not accelerate the acute renal failure. Instead, the accelerated tubular damage due to HS appears to have worsened the uremic state associated with AKI, as the serum level of indoxyl sulfate (IS), a representative uremic toxin excreted from proximal renal tubules [38–42], was significantly higher in HS-fed IR mice than in IR mice without HS (Fig 2B). Thus, the severe uremic state associated with AKI might be a major cause of the high mortality in HS-fed IR mice. Consistent with the worse health condition in mice due to uremia, food and water intake were reduced in HS-fed IR mice compared with IR mice; accordingly, the decrease in bodyweight was greater in HS-IR mice (Fig 2C). Such states might further increase mortality in HS-fed IR mice. Note that no reduction in food/water intake was observed in non-IR sham mice under HS (Fig 2C).

Serum chloride levels were significantly higher in IR mice with either an HS diet or injection of 0.9% NaCl solution than those without HS (Fig 2D). Thus, metabolic acidosis due to the high serum level of chloride might further increase the mortality of HS-fed IR mice. Because non-IR sham mice did not exhibit hyperchloremia under HS (Fig 2D), it is likely that the increase in serum chloride levels under HS loading was caused by deficient tubular functions due to the IR-mediated proximal tubular damage. Indeed, urinary chloride levels were also higher in HS-IR than IR mice, while they were low, comparable to those in IR mice in non-IR sham mice with HS (Fig 2D). Accordingly, serum sodium levels were also higher in IR mice under HS (Fig 2E). In contrast, serum potassium levels were comparable in all types of mice (Fig 2F).

It is also likely that HS caused dehydration, which also resulted in hyperchrolemia and the increase in mortality. This is common in particular in elderly patients with AKI [43]. To test this idea, we addressed the serum levels of total protein and albumin in IR mice with or without HS, and HS-sham mice. As shown in Fig 2G, their serum levels were almost comparable till day 3 after IR, suggesting that the dehydration was not the maijor cause for the high mortality observed in HS-IR mice.

Because there was no significant difference in renal blood flow (S2A Fig) or systolic blood pressure (S2B Fig) between HS-fed IR mice and IR mice without HS, increased vasoconstriction might not be involved in the exacerbation of AKI by HS. In addition, the mRNA levels of pro-inflammatory factors were assessed by QPCR using kidney RNA (Fig 2H), and moreover, their protein levels were also addressed in F4/80⁺ macrophages isolated from the kidney by

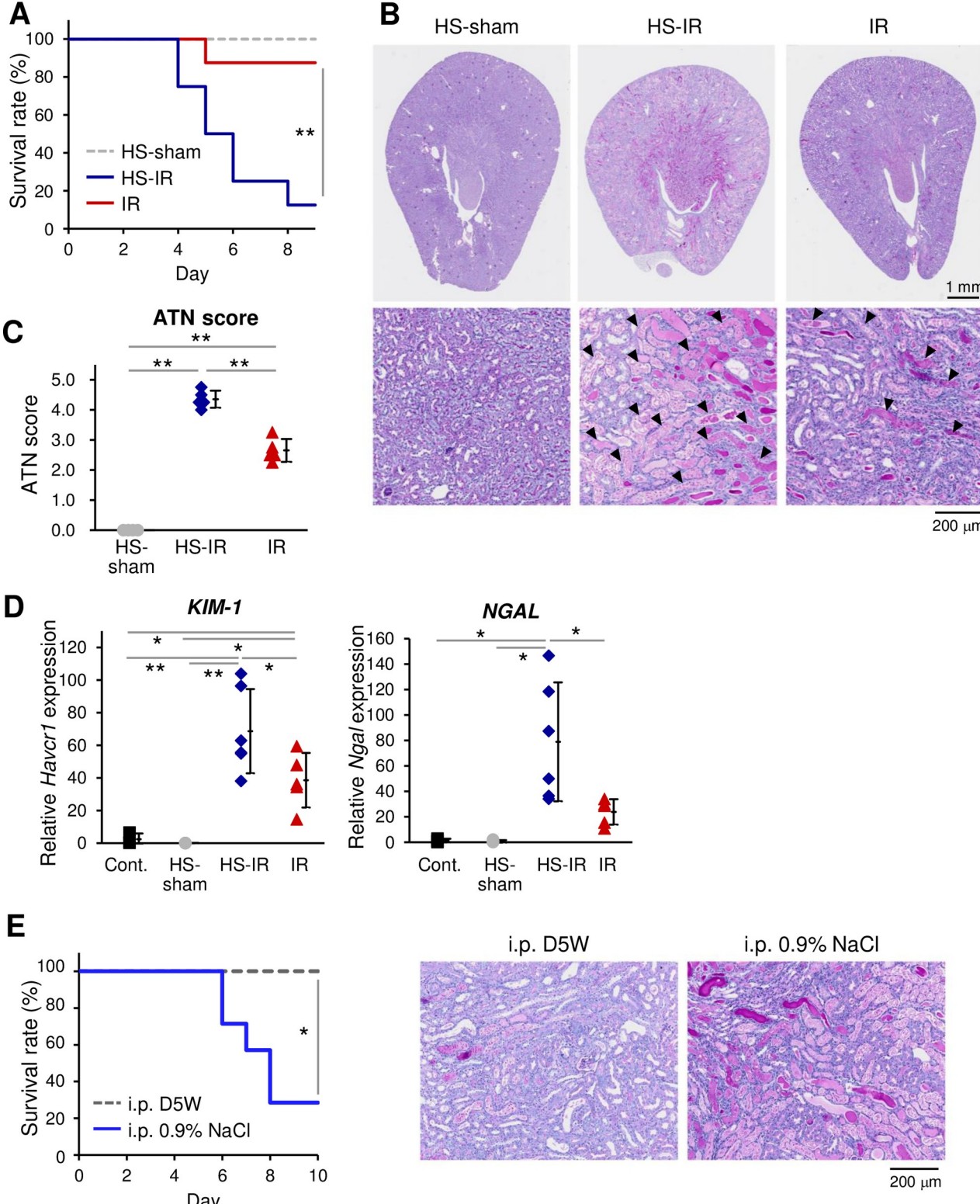

**Fig 1. Detrimental effect of high salt on IR-induced AKI. (A) Survival curves.** Kaplan-Meier survival curves of wild-type AKI mice with (HS-IR) or without (IR) HS (*n* = 8 per group). The significance of differences between HS-IR and IR was calculated with the log-rank test. Sham-treated mice (without IR) fed an HS diet were also analyzed (HS-sham, *n* = 4). **(B) Representative PAS-stained histological images.** Presented images are of the whole kidney and corticomedullary junction area from HS-sham (*n* = 4) and IR mice with (HS-IR) or without (IR) HS (*n* = 5 per group) on day 3

after IR/sham. Black arrowheads indicate intraluminal debris. **(C) ATN scores on day 3 after IR/sham.** $n$ = 4 for HS-sham and $n$ = 5 for IR mice with or without HS. Scales are presented in the figure. Statistics: one-way ANOVA followed by Holm's *post hoc* test. **(D) QPCR analysis**. Values of the mRNA expression of the kidney injury markers *Haver1* (*KIM-1*) and *Ngal* in the kidney of HS-sham ($n$ = 4), HS-IR ($n$ = 6), and IR ($n$ = 5) mice on day 3 after IR/sham are presented. The values from mice without any treatment (Cont.) are also presented ($n$ = 3). Statistics: one-way ANOVA followed by Holm's *post hoc* test. Note that we used different set of mice for histologic analysis (C) and mRNA analysis (E), as each analysis requires its own pre-treatments of the tissues. **(E) Effects of intraperitoneal injection of 0.9% NaCl.** (*left*) Kaplan-Meier survival curves ($n$ = 4 per group) of mice with daily intraperitoneal injection of D5W or 0.9% NaCl injection after IR/sham. (*right*) Representative PAS-stained histological images of the corticomedullary junction area from mice injected with D5W or 0.9% NaCl.

flowcytometry (S3A Fig). No difference was observed between IR and HS-fed IR mice, which denied the possibility that HS advanced interstitial inflammation during AKI progression [44–47]. Similarly, the mRNA levels of fibrotic genes were also comparable in IR and HS-fed IR mice (S3B Fig).

## AIM ameliorates HS-induced lethal AKI

We wondered why HS accelerated the tubular damage. We previously reported that the reduced dissociation of AIM from IgM pentamer in serum during AKI fails to facilitate the removal of intraluminal debris, resulting in exacerbation of tubular injury [27]. Typically, feline AIM is unable to dissociate from IgM pentamer due to its extremely high binding affinity to the IgM-Fc region, which results in a markedly high susceptibility to kidney disease in cats [48]. Because the accelerated tubular damage and high mortality observed in HS-IR mice were reminiscent of the phenotype in AIM-deficient ($AIM^{-/-}$) mice [27] or cats [48] after IR, the HS may have decreased the dissociation of AIM from IgM, thereby exacerbating the renal tubular injury.

To test this hypothesis, we assessed the state of AIM release from IgM pentamer in the serum of HS-IR mice. The levels of IgM-free AIM were significantly lower in HS-fed IR mice than in IR mice, particularly on day 1 (Fig 3A; whole blots are presented in S4 Fig). In addition, the amount of urinary AIM was also lower in HS-fed IR mice than in IR mice on day 1 when assessed using daily deposited urine (Fig 3B). Hence, it is likely that HS interfered with the dissociation of AIM from IgM pentamer, resulting in the acceleration of AKI with a further exacerbation of luminal obstruction.

In accordance with this result, $AIM^{-/-}$ CD1 mice, which were newly created using the CRISPR/Cas9 strategy, exhibited worse survival predominantly during the acute phase (days 2 and 3) than wild-type CD1 mice in response to IR plus HS, whereas the difference in $AIM^{-/-}$ CD1 mice under IR with or without HS was less significant, clearly indicating the involvement of AIM in the difference in AKI severity between HS-IR and IR mice (Fig 3C).

We then tested the therapeutic effect of the administration of recombinant AIM (rAIM) on such lethal AKI under HS. Impressively, daily injection of rAIM (200 μg/mouse) from day 1 after IR dramatically improved the survival of HS-fed IR wild-type mice: rAIM injection increased overall survival at day 7 to 86% from 17% in HS-fed IR mice (Fig 3D). The loss of body weight during days 2 and 3 was also lower in rAIM-injected mice, in conjunction with the improved health state (Fig 3E). Histologic analysis showed that luminal dead cell debris was markedly decreased with rAIM injection (Fig 3F), which was in parallel with the improvement of the ATN score (Fig 3G). Quantitative PCR analysis revealed that rAIM administration decreased *Havcr1* (*KIM-1*) and *Ngal* mRNA levels in the kidney, indicating an improvement of tubular damage with rAIM (Fig 3H). Serum IS levels were also reduced significantly by rAIM injection, implying an improvement in the uremic state (Fig 3I). Similarly, rAIM injection improved survival and histology in IR wild-type mice with intraperitoneal infusion of 0.9% NaCl (Fig 3J). Altogether, the therapeutic effect of rAIM administration was clear, even for lethal AKI. Such a pronounced effect of rAIM is in accordance with the results (Fig 3A)

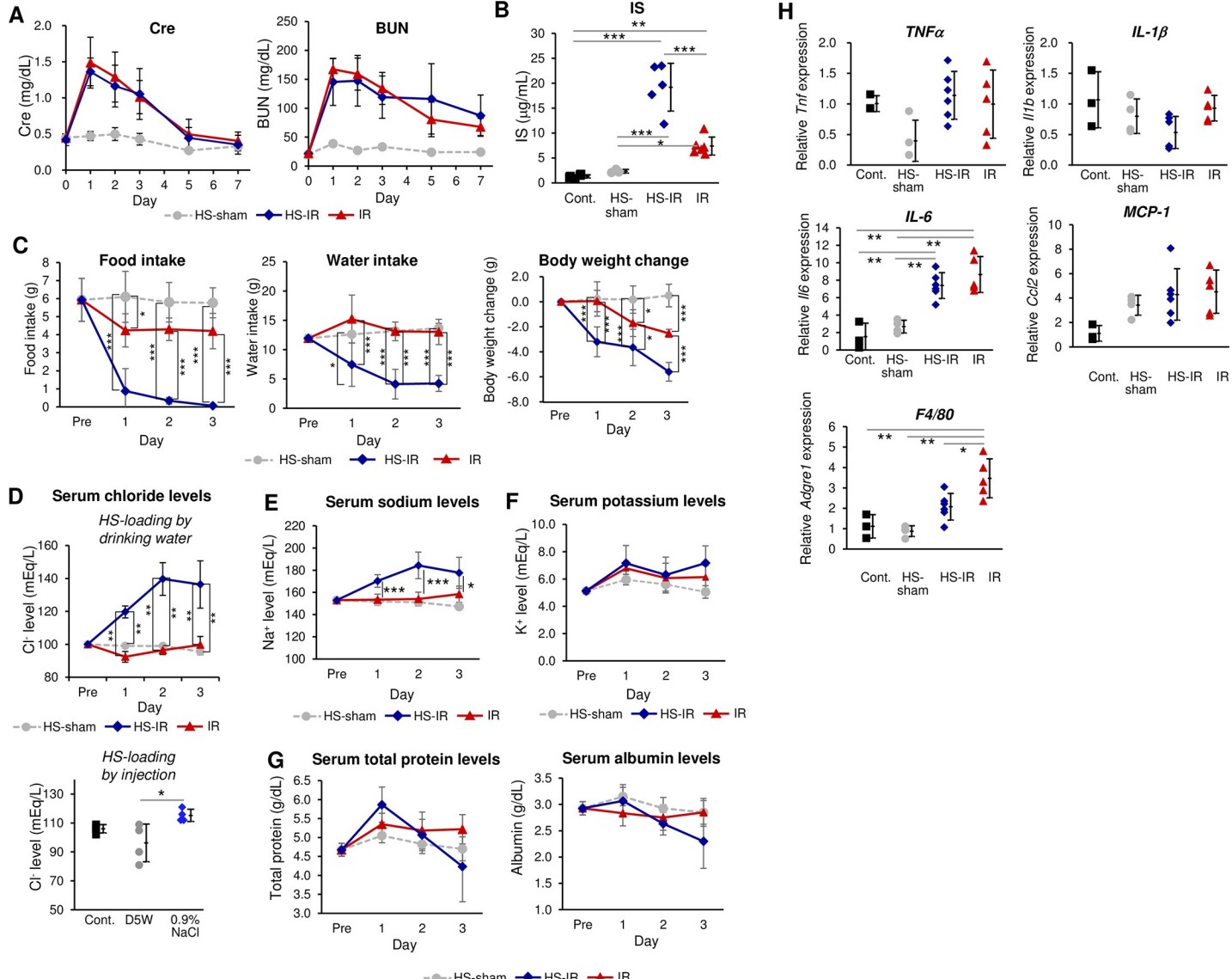

**Fig 2. How high salt exacerbates AKI. (A) Serum Cre and BUN levels after IR.** HS-sham (*n* = 4), HS-IR (*n* = 6), and IR (*n* = 5) mice were analyzed. No significant differences between HS-IR and IR were detected at any time point with Welch's *t*-test. **(B) Serum IS levels on day 3 after IR/sham.** HS-IR (*n* = 5) and IR (*n* = 6) mice were analyzed. Statistics: Multi-way ANOVA with Bonferroni's adjustment. **(C) Food intake, water intake, and body weights.** Values relative to those before IR in HS-sham (*n* = 4), IR (*n* = 5), and HS-IR (*n* = 6) mice are presented. Statistics: Welch's *t*-test with Holm's adjustment. **D. Serum chloride levels.** Results of HS-sham (*n* = 4), IR (*n* = 5), and HS-IR (*n* = 6) (upper panel), results of IR mice injected with D5W or 0.9% NaCl (*n* = 4 each) at day 3 after IR/sham and in control mice (*n* = 3) (lower panel). Statistics: multi-way ANOVA followed by Bonferroni's *post hoc* test (upper panel). One-way ANOVA followed by Holm's *post hoc* test (lower panel). **(E) Serum sodium and (F) potassium levels.** Values after IR. HS-sham (*n* = 4), HS-IR (*n* = 6), and IR (*n* = 5) mice were analyzed. Statistics: Welch's t-test with Holm's adjustment. *: HS-IR v. s. IR. **G: Serum levels of total protein and albumin.** Sera from HS-sham (*n* = 4), IR (*n* = 6), and HS-IR (*n* = 6) ICR mice were analyzed on day 1 to day 3 after IR. Values were also tested in ICR mice without IR or HS-fed as "Pre" (n = 4). Statistics: Welch's *t*-test with Holm's adjustment. **H. QPCR analysis.** mRNA expression of various pro-inflammatory genes in HS-sham (*n* = 4), IR (*n* = 5), and HS-IR (*n* = 6) mice on day 3 after IR/sham. The values from the mice without any treatment (Cont.) are also presented (*n* = 3). Statistics: one-way ANOVA followed by Holm's *post hoc* test.

suggesting that the reduction of serum IgM-free AIM levels was a major reason for the exacerbation of AKI in HS-fed IR mice.

In contrast, the high serum chloride (and sodium) levels were comparable between HS-fed IR mice with or without rAIM injection (Fig 3K), suggesting that AIM did not improve the metabolic acidosis caused by hyperchloremia. Based on this result, it is unlikely that metabolic acidosis was a major cause of the mouse mortality under HS loading.

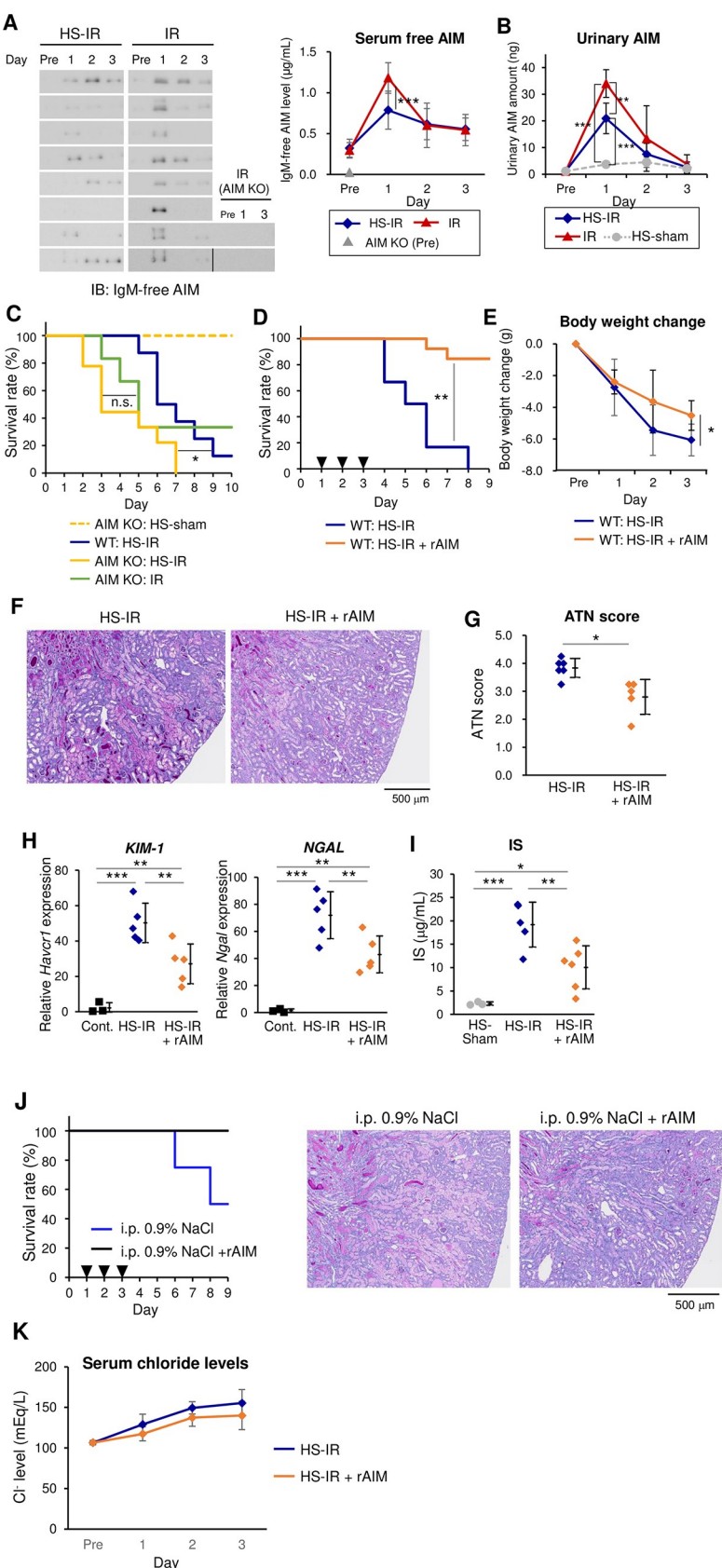

**Fig 3. Therapeutic effects of AIM on lethal AKI. (A) Immunoblotting for serum AIM.** Immunoblotting under non-reducing conditions was performed using rabbit anti-mouse AIM polyclonal antibody (rab2, made in-house). Immunoblots for IgM-free AIM, detected at 37 kDa [27], are cropped from the full-length images shown in S4 Fig. The cropped images from different fields of same or different gels are separated by spaces or a dividing line. The blot for the serum from AIM KO mice ($n$ = 2) were also presented as negative controls. The concentration of IgM-free AIM determined on the indicated days after IR using NIH ImageJ and are presented in the right panel. The value in AIM KO mice at Pre is also presented as a negative control (gray triangle). IR mice, $n$ = 8; and HS-IR mice, $n$ = 8. Statistics: Muti-way ANOVA followed by Bonferroni's *post hoc* test. **(B) Urinary AIM levels.** AIM levels were assessed on the indicated days after IR by ELISA. IR mice, $n$ = 3; and HS-IR mice, $n$ = 4. Statistics: Muti-way ANOVA followed by Bonferroni's *post hoc* test. **C. Kaplan-Meier survival curves of HS-fed wild-type (WT, $n$ = 8) and $AIM^{-/-}$ (AIM KO, $n$ = 9) mice.** Statistics: log-rank test. The survival of HS-sham $AIM^{-/-}$ mice ($n$ = 3) and IR-treated $AIM^{-/-}$ mice ($n$ = 6) are also presented. Statistics: log-rank test. **D. Kaplan-Meier survival curves.** HS-IR wild-type mice with ($n$ = 13) or without ($n$ = 6) daily intravenous injection of rAIM (200 μg). Arrowheads indicate the times of rAIM injection. Statistics: log-rank test. **E. Body weights.** Values relative to pre-IR in HS-IR mice with ($n$ = 6) and without ($n$ = 5) rAIM. Means ± s. d. are shown. Statistics: Welch's t-test with Holm's adjustment. ** $P$ < 0.01. **F. Representative PAS-stained histological images.** The kidney including the corticomedullary junction area on day 3 after IR from HS-IR mice with ($n$ = 5) or without ($n$ = 6) rAIM injection. Scale bar, 500 μm. **G. ATN scores on day 3.** HS-IR mice with ($n$ = 5) or without ($n$ = 6) rAIM injection. Statistics: Welch's t-test. **H. QPCR analysis.** *Havcr1* (*KIM-1*) and *Ngal* mRNA levels in the kidney on day 3 in HS-IR mice with ($n$ = 5) or without ($n$ = 5) rAIM injection. The values from the mice without any treatment (Cont.) are also presented ($n$ = 3). Statistics: one-way ANOVA followed by Holm's *post hoc* test. Note that we used different set of mice for histologic analysis (E, F) and mRNA analysis (G), as each analysis requires its own pre-treatments of the tissues. **I. Serum IS levels**. Values on day 3 after IR. HS-IR mice with ($n$ = 6) and without ($n$ = 5) rAIM injection, HS-sham ($n$ = 3) were analyzed. Statistics: one-way ANOVA followed by Holm's *post hoc* test. **J. Survival curves (left) and representative photos of PAS-stained kidney specimens (right).** IR mice subjected to i. p. injection of 0.9% NaCl with or without rAIM i.v. injection (n = 2 each) on day 3 after IR. **K. Serum chloride levels.** Values were assessed at indicated days in HS-IR mice with ($n$ = 6) and without ($n$ = 5) rAIM injection. Means ± s. d. are shown. No significance between with and without rAIM injection was detected by Welch's t-test.

## AIM is ineffective for lethal AKI in the absence of KIM-1

We previously demonstrated that surviving tubular epithelial cells have a role in the phagocytic removal of intraluminal dead cell debris via KIM-1 expressed on the luminal cell surface, leading to recovery from AKI [27]. As a ligand of KIM-1, AIM accumulation on the debris is indispensable for promoting the efficient engulfment of the debris.

To determine whether the AIM-KIM-1 axis is also responsible for the repair of lethal AKI induced by IR plus HS loading, newly generated KIM-1-deficient ($KIM-1^{-/-}$) mice received IR surgery, were fed the HS. The survival of $KIM-1^{-/-}$ mice was worse when fed HS (Fig 4A), though the difference was not as profound as observed in wild-type mice (presented in Fig 1A). We then tested the therapeutic effect of rAIM in $KIM-1^{-/-}$ mice under IR plus HS. As shown in Fig 4A, although rAIM injection improved the survival of $KIM-1^{-/-}$ mice subtly but significantly, the effect was not as profound as observed in wild-type mice (presented in Fig 3D). In accordance with these findings, in $KIM-1^{-/-}$ mice, histologic analysis revealed a large amount of luminal debris with or without rAIM injection (Fig 4B) and the ATN score was not improved by rAIM injection (Fig 4C). Serum IS levels were not reduced in $KIM-1^{-/-}$ mice by rAIM (Fig 4D).

All these results indicate that the repair of HS-induced lethal AKI by rAIM was largely achieved through the KIM-1-mediated phagocytic removal of intraluminal debris. However, the worse survival in IR-$KIM-1^{-/-}$ mice under HS and the subtle improvement of the survival of the same mice by rAIM injection suggest that there might be minor additional rAIM effects, which improved the survival of mice suffering severe AKI independently of KIM-1.

## Discussion

This study provides several new insights into the effect of salt on AKI. Firstly, using animal models, we demonstrated that high salt loading accelerated proximal tubular injury, leading to lethal AKI. Most renal tubules were obstructed by a large amount of dead epithelial cell debris,

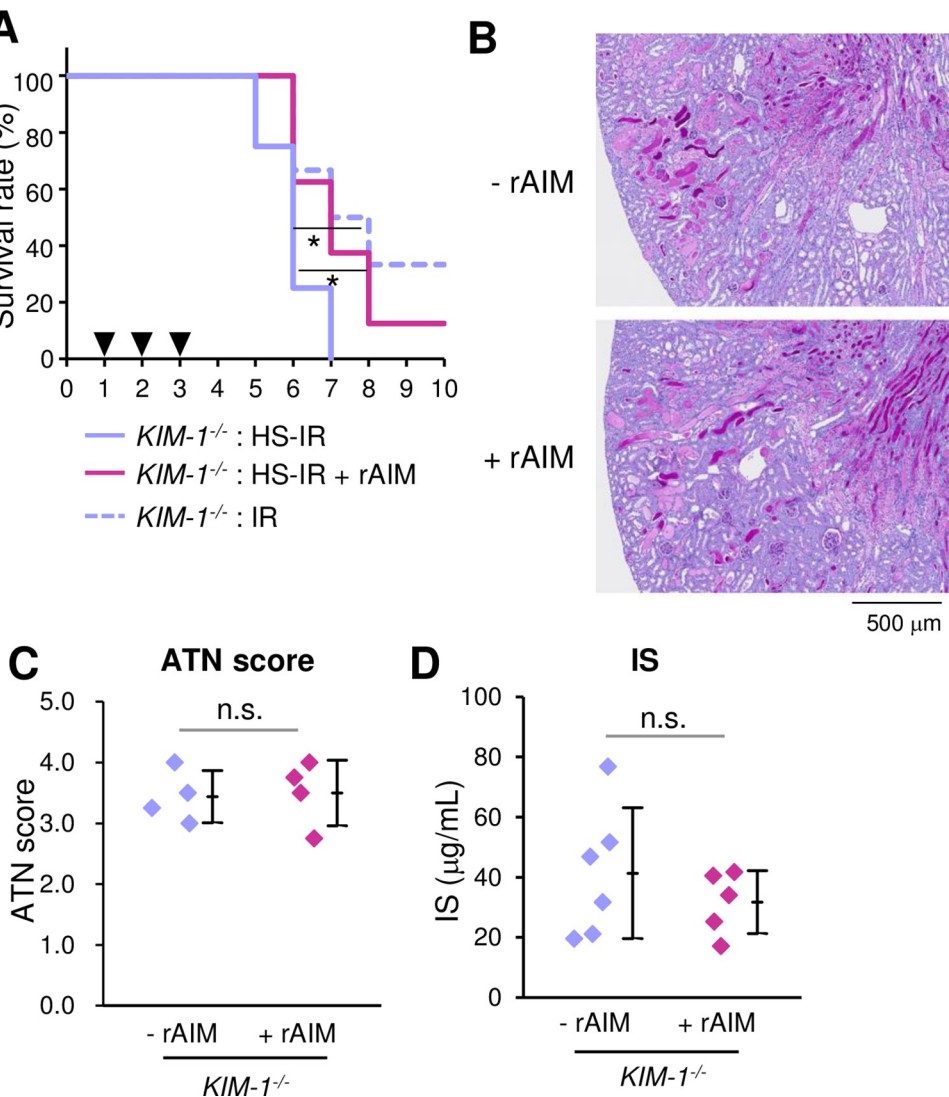

**Fig 4. Involvement of the AIM-KIM-1 axis in the repair process of lethal AKI. (A) Kaplan-Meier survival curves**.
Wild-type and HS-IR *KIM-1*$^{-/-}$ mice with or without rAIM injection (200 μg). *n* = 8 for *KIM-1*$^{-/-}$ mice; and *n* = 8 for
rAIM-injected *KIM-1*$^{-/-}$ mice. IR without HS (*n* = 6) is also presented. Arrowheads indicate the times of rAIM
injection. Statistics: log-rank test. **(B) Representative photomicrographs of PAS-stained kidney specimens**. The
kidney images including the corticomedullary junction area from wild-type (*KIM-1*$^{+/+}$) mice and HS-IR *KIM-1*$^{-/-}$
mice with or without rAIM injection on day 3 after IR (*n* = 3 each) are presented. **(C) ATN scores on day 3 after IR.** *n*
= 4 each. Statistics: Welch's t-test. **(D) Serum IS levels.** Values on day 3 after HS-IR. *n* = 6 for *KIM-1*$^{-/-}$ mice; and *n* =
5 for rAIM-injected *KIM-1*$^{-/-}$ mice. IR without HS (*n* = 6) are presented. Statistics: Welch's t-test.

which resulted in a whitened macroscopic appearance of the kidney in HS-fed IR mice. Such
histologic and macroscopic views were well associated with significant increases in the serum
levels of different biomarkers representative of tubular damage. There was no significant dif-
ference in serum Cre/BUN levels between HS-fed IR and IR mice, suggesting that the detri-
mental action of salt was preferentially on renal tubules rather than on glomeruli. Thus, the
mouse mortality increase induced by HS appeared mostly due to the possible exacerbation of
the uremic state evident by the increase in the serum IS level, and the resulting worse general
health state. This is possible because types of uremic toxins in addition to IS are excreted from
proximal tubules [40, 49–54] and therefore, severe tubular damage caused by HS-IR should

result in their accumulation leading to lethal uremia. Such a rapid impact of HS during the acute progression of AKI was surprising as it is believed that salt affects kidney function chronically.

In this report, we employed newly generated $AIM^{-/-}$ CD1 mice, as it was reported that this strain is sensitive to IR [37]. Although little is known about renal blood flow distribution in CD1 mice, there could be certain effects by the HS on renal medullary blood flow specifically in CD1 mice such as the occlusion of vessels due to a lack of recovery following the ischemia. This issue need to be further studied.

Secondly, we found a new mechanism that HS loading interfered with the dissociation of AIM from IgM pentamer, which resulted in the less efficient removal of luminal debris, thereby further exacerbating AKI. The striking therapeutic effect of rAIM injection in HS-fed IR mice may support this mechanism.

In serum, AIM is released from IgM pentamer in different diseases, including AKI [27, 48]. However, the precise physiological mechanism promoting AIM dissociation remains unknown, and thus it is difficult to specify how HS loading suppresses the dissociation of AIM. In the clinical setting, different groups have claimed that hyperchloremia is a risk factor for AKI and is often caused by the administration of fluids containing super-physiological chloride concentrations [44, 55–58]. It is noteworthy that 0.9% saline, so-called "normal saline", also fulfills this criterion [59, 60]. Indeed, Bailey's group recommended a chloride-restrictive strategy, showing a significant decrease in the incidence of AKI by this strategy in a tertiary intensive care unit [57]. Despite the obvious clinical association between hyperchloremia and AKI, the underlying mechanisms that link the two issues remain unknown. Considering all these observations, a hypothesis emerges that hyperchloremia might be associated with AKI by disturbing the dissociation of AIM from the IgM pentamer. Clinical studies focusing on IgM-free AIM levels in AKI patients receiving hyper-chloride fluids are required to examine this idea.

On the other hand, the vasoconstriction caused by tubuloglomerular feedback due to the delivery of high-dose chloride to the macula densa may be the most widely accepted mechanism to explain the detrimental action of hyperchloremia on the progression of AKI [44, 61–65]. Vasoconstriction results in a reduction of the glomerular filtration rate and can also be applied to the pathophysiological mechanism of how salt worsens CKD [44, 61–65]. However, this was not the case in HS-fed IR mice, as their body weight decreased and there were no differences in renal blood flow and systolic blood pressure compared with IR mice without the HS. Probably, the disrupted reabsorption of water due to the IR-induced proximal tubular injury might have inhibited the effect of vasoconstriction [65]. Otherwise, more direct and body fluid-independent effects of high salt loading on renal function have been proposed [66]. Yu et al. showed that dietary salt promoted widespread glomerular and tubular fibrosis [67]. It was demonstrated later that the promotion of renal fibrosis by high salt loading is associated with augmented transforming growth factor-β1 production in the kidney. It has also been reported that high salt loading increases the production of reactive oxygen species, which stimulate renal fibrosis and renal cell death [68]. In addition, various animal studies have shown that long-term challenge with high salt loading decreases the number of microcirculatory vessels in the kidney, leading to a reduction in renal tissue oxygenation [69–71]. Furthermore, recent evidence has suggested that high salt loading influences kidney function through different immunological pathways, including the induction of low-grade, chronic inflammation in the kidney in humans and animal models [13]. Moreover, Matthias et al. reported that salt increases the number of pathogenic Th17 cells, which injure kidney tissue [72]. However, all these mechanisms induce a mostly chronic effect and, therefore, must be associated with CKD, and not with AKI. Actually, the duration of AKI in the present study (<7 days) was too short

to complete microvascular remodeling or renal fibrosis. Indeed, quantitative PCR analysis indicated no significant differences in the mRNA expression of inflammatory and fibrotic genes in IR mice with or without the HS. Thus, at least for the exacerbation of tubular injury, the detrimental effect of HS loading appears to be mainly brought about by a decrease of AIM dissociation from IgM pentamer.

Lastly, we were surprised to observe that rAIM administration a strong effect of rescuing mice from lethal AKI induced by IR and HS loading. It is likely that such a strong therapeutic effect of AIM against AKI was mainly achieved through the removal of tubular-obstructing dead cell debris using the AIM-KIM-1 axis, as the lethal AKI induced in *KIM-1*$^{-/-}$ mice by IR plus HS was not improved by rAIM administration as profoundly as in wild-type mice. Intriguingly, however, the survival of *KIM-1*$^{-/-}$ mice after IR was subtly, but significantly, worse under HS fed (Fig 4A). In addition, rAIM injection in the same mice improved their survival subtly, though the state of kidney injury was not altered (Fig 4A). These results imply the presence of additional beneficial effects of AIM on the survival of IR-treated mice under HS, which is independent of KIM-1. Further study will be required to clarify the unknown effects.

In conclusion, our current study corroborated the detrimental effect of NaCl on AKI and dissected its pathological process. In addition, we demonstrated the potent therapeutic effect of AIM on lethal AKI. These findings could be the basis for a new paradigm of AKI control in human patients.

## Supporting information

**S1 Fig. The validation of the newly generated *AIM*$^{-/-}$ CD1 mice and the *KIM-1*$^{-/-}$ mice.** **(A)** Serum from newly generated *AIM*$^{-/-}$ and *AIM*$^{+/+}$ CD1 littermates were analyzed by immunoblotting in non-reducing condition using anti-AIM antibody (*n* = 3 each). No AIM protein was detected in *AIM*$^{-/-}$ serum. **(B)** Representative photos of immunohistochemistry for KIM-1 on day 3 after HS-IR in newly generated *KIM-1*$^{-/-}$ and the littermates *KIM-1*$^{+/+}$ are shown (n = 3). Control (cont. without any treat) is also presented. No KIM-1 expression was induced after HS-IR in *KIM-1*$^{-/-}$ mice while KIM-1 was highly expressed in the corticomedullary junction area in *KIM-1*$^{+/+}$ mice after HS-IR.
(PDF)

**S2 Fig. Renal blood flow and systolic blood pressure changes after HS-IR. (A)** Percentage decrease in total renal blood flow after IR in IR (*n* = 8) and HS-IR (*n* = 9) mice. **(B)** Systolic blood pressure in IR (*n* = 5) and HS-IR (*n* = 6) mice on the indicated days after IR. No significant differences between HS-IR and IR were detected at any time point with Welch's t-test.
(PDF)

**S3 Fig. Changes in pro-inflammatory and fibrotic factors after HS-IR. (A)** F4/80-high or low cells in CD45$^{+}$Ly-6G$^{-}$CD11b$^{+}$ cells prepared from kidneys (indicated in the upper panels) were analyzed for the intracellular expression of various pro-inflammatory factors in HS-IR and IR (*n* = 3 each) mice on day 3 after IR/sham. The results from the mice without any treatment (Cont.) are also presented (*n* = 3). Statistics: one-way ANOVA followed by Holm's *post hoc* test. **(B)** QPCR analysis of the mRNA expression of fibrotic genes performed as in Fig 2H.
(PDF)

**S4 Fig. Full-length images of immunoblots for serum IgM-free AIM.** Full-length images of immunoblots for serum IgM-free AIM presented in Fig 3A are shown. Immunoblotting for serum AIM together with recombinant AIM (rAIM) in non-reducing was performed using rabbit anti-mouse AIM polyclonal antibody (rab2, made in-house). After IR, IgM-free AIM at

37 kDa (surrounded by red squares) was increased in the IR group (n = 8) especially on day1 (D1) while no or little IgM-free AIM was detected in the "Pre" condition. In contrast, in the HS-loaded condition (HS-IR group, n = 8), the levels of IgM-free AIM after IR were lower than those in the IR group. IgM-bound AIM and backgrounds are surrounded by blue and green squares, respectively. The serum from AIM KO mice are also presented as negative controls besides HS-IR#6, 8 and IR#7, 8. Calculated IgM-free AIM concentration (μg/mL) is presented under each band.
(PDF)

**S1 Table. qPCR primer list.**
(PDF)

## Acknowledgments

We thank Drs. T. Shimozawa, K. Yasuda and K. Kurokawa for useful discussion and advise, and A. Nishijima and A. Hirota for general technical assistance.

## Author Contributions

**Data curation:** Satoko Arai, Toru Miyazaki.

**Formal analysis:** Satoko Arai, Toru Miyazaki.

**Funding acquisition:** Satoko Arai, Toru Miyazaki.

**Investigation:** Ching-Ting Wang, Tetsushi Tezuka.

**Resources:** Naoki Takeda, Kimi Araki.

**Supervision:** Toru Miyazaki.

**Validation:** Satoko Arai, Toru Miyazaki.

**Visualization:** Satoko Arai.

**Writing – original draft:** Toru Miyazaki.

**Writing – review & editing:** Satoko Arai, Toru Miyazaki.

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
