## [Decision Letter · Decision Letter 0]

23 Sep 2021

PONE-D-21-26872High salt exacerbates acute kidney injury by disturbing the activation of CD5L/apoptosis inhibitor of macrophage (AIM) proteinPLOS ONE

Dear Dr. Miyazaki,

Thank you for submitting your manuscript to PLOS ONE. After careful consideration, we feel that it has merit but does not fully meet PLOS ONE’s publication criteria as it currently stands. Therefore, we invite you to submit a revised version of the manuscript that addresses the points raised during the review process. Two reviewers have read your manuscript. Both find it of interest, comprehensive and well-written. They had practically no comments (see below). Note that one of them failed to successfully submit comments via the dashboard but I received a mail from this reviewer. I have one comment that needs your attention. The huge increase in serum chloride and sodium can only be due to extreme hyperosmolar dehydration secondary to the failure of these very ill mice to maintain fluid intake and possibly polyuria. I also think that this very severe dehydration lead to the very high mortality rather than the accumulation of uremic toxins that typically can be tolerated for longer periods. This is quite common in elderly patients with AKI, see [Hyperosmolar dehydration: A predictor of kidney injury and outcome in hopitalized older adults. El-Sharkawy et al. Clin Nutr. 2020;39:2593-9]. This study should be cited. Maintenance of normal serum urea, creatinine and potassium also suggests that renal function, although impaired as suggested by increased IS, was not yet at end-stage kidney disease. Did you measure hematocrit? If so, please include these data. I suspect a marked increase. If not, please measure total protein or albumin in serum. These are also proxy markers of dehydration (although less accurate than hematocrit). If no serum was saved then at least calculate serum osmolality with the standard simple formula for normoglycemic uremic serum (serum osmolality = 2*Na + 2*K +urea). If you measured urine production/24h please provide these data as well. Urine osmolality would also be useful. Please move the serum Na and serum K data (currently Figure S2) to the main body.  Finally, adapt the results and discussion to incorporate these insights.

We look forward to receiving your revised manuscript.

Kind regards,

Jaap A. Joles, DVM, PhD

Academic Editor

PLOS ONE

Journal Requirements:

[We thank Drs. T. Shimozawa, K. Yasuda and K. Kurokawa for useful discussion and advise, and A. Nishijima and A. Hirota for general technical assistance. This work was supported by AMED-CREST and AMED-LEAP, Japan Agency for Medical Research Development (to T.M.), and MEXT Grant-in-Aid for Scientific Research (B) grant number 16H05313 (to S.A.).]

 [This work was supported by AMED-CREST and AMED-LEAP, Japan Agency for Medical Research Development (to T.M.), and MEXT Grant-in-Aid for Scientific Research (B) grant number 16H05313 (to S.A.).]

Reviewers' comments:

Reviewer's Responses to Questions

**Comments to the Author**

1. Is the manuscript technically sound, and do the data support the conclusions?

Reviewer #1: Yes

2. Has the statistical analysis been performed appropriately and rigorously? 

Reviewer #1: Yes

3. Have the authors made all data underlying the findings in their manuscript fully available?

Reviewer #1: Yes

4. Is the manuscript presented in an intelligible fashion and written in standard English?

Reviewer #1: Yes

5. Review Comments to the Author

Reviewer #1: In this manuscript, the authors present their recent data on the role of AIM/CD5L in the setting of salt-induced exacerbation of AKI. Using elegant series experiments with AIM KO and KIM-1 KO mice in combination with purified preparations of AIM, the authors showed that high salt blocks the release of AIM leading to a blockage of the phagocytotic removal of intraluminal cellular debris. Presentation is logical based on solid series of data and the major conclusions are supported by the data presented. This manuscript provides novel insight into the role of AIM-KIM-1 pathway in salt-induced worsening of AKI independent of metabolic acidosis.

I have one minor editorial comment: In page 16, line 2, “urine” should be “serum.”

6. PLOS authors have the option to publish the peer review history of their article (what does this mean?). If published, this will include your full peer review and any attached files.

Reviewer #1: **Yes: **Hyug Moo Kwon

---

## [Author Response · Author response to Decision Letter 0]

6 Nov 2021

Response. Thank you for the reviewer’s comment. In response to the comment, we addressed the serum levels of total protein and albumin in IR mice with or without high-salt, as well as in sham mice with high-salt to test the presence of dehydration in mice fed with high-salt. As demonstrated in the new Fig 2G, the levels of both items were almost comparable on day 1 to day 3 after IR. Hence, it is unlikely that severe dehydration was the major cause of high mortality and hyperchloremia observed in IR mice fed with high-salt. We described this observation and conclusion in page 16 line 15, also citing the indicated paper (reference 43). In addition, as the reviewer suggested, we have moved the S2 Fig to the main body as Fig 2E and 2F.

---

## [Editor Report · Decision Letter 1]

10 Nov 2021

High salt exacerbates acute kidney injury by disturbing the activation of CD5L/apoptosis inhibitor of macrophage (AIM) protein

PONE-D-21-26872R1

Dear Dr. Miyazaki,

We’re pleased to inform you that your manuscript has been judged scientifically suitable for publication and will be formally accepted for publication once it meets all outstanding technical requirements.

Kind regards,

Jaap A. Joles, DVM, PhD

Academic Editor

PLOS ONE
---

## [Editor Report · Acceptance letter]

12 Nov 2021

PONE-D-21-26872R1 

High salt exacerbates acute kidney injury by disturbing the activation of CD5L/apoptosis inhibitor of macrophage (AIM) protein 

Dear Dr. Miyazaki:

I'm pleased to inform you that your manuscript has been deemed suitable for publication in PLOS ONE. Congratulations! Your manuscript is now with our production department. 

Kind regards, 

on behalf of

Dr. Jaap A. Joles 

Academic Editor

PLOS ONE